# Mineral and Metabolome Analyses Provide Insights into the Cork Spot Disorder on 'Akizuki' Pear Fruit

Yingjie Yang [1,2,†], Yanlin Zhu [1,†], Piyu Ji [1], Anqi Li [1], Zhiyun Qiu [1], Yuanyuan Cheng [1], Ran Wang [1], Chunhui Ma [1], Jiankun Song [1], Zhenhua Cui [1], Jianlong Liu [1], Yitian Nie [3], Xiaozhi Zhou [4] and Dingli Li [1,*]

[1] College of Horticulture, Qingdao Key Lab of Genetic Improvement and Breeding of Horticultural Plants, Qingdao Agricultural University, Qingdao 266109, China; wuhuaguoyyj@163.com (Y.Y.); zylqau@163.com (Y.Z.); jipiyu123@163.com (P.J.); 17806267627@163.com (A.L.); qzyqau@126.com (Z.Q.); chengyy0923@163.com (Y.C.); qauwr@126.com (R.W.); machunhui2000@163.com (C.M.); qausjk@126.com (J.S.); zhcui@qau.edu.cn (Z.C.); 201901068@qau.edu.cn (J.L.)

[2] Academy of Dongying Efficient Agricultural Technology and Industry on Saline and Alkaline Land in Collaboration with Qingdao Agricultural University, Dongying 257399, China

[3] Qingdao Onatur Agri-Tech Co., Ltd., Pingdu, Qingdao 266700, China; fieldernie@163.com

[4] Department of Finance, Qingdao Agricultural University, Qingdao 266109, China; zhouxiaozhi06@126.com

\* Correspondence: lidingli@qau.edu.cn

† These authors contributed equally to this work.

**Abstract:** Cork spot is a common physiological disorder in pear fruits, which has been found in some pear cultivars. Mineral nutrition imbalance in fruit is regarded as the principal influence factor for disorder incidence, with some ongoing confusion and controversy. In our research, we explored the cork spot characteristics in Japanese pear 'Akizuki' (*Pyrus pyrifolia* Nakai), adopted metabolome and mineral content analysis for healthy and disordered fruits, and made a correlation analysis of mineral and metabolites. Cork spots are mainly distributed on the outer flesh beneath the fruit peel. In cork spotted tissues, superoxide (SOD) and peroxidase (POD) activities, as well as malondialdehyde (MDA) content, increased. A total of 1024 known metabolites were identified from all the samples and more changes in metabolism were detected between normal and cork spotted flesh tissues. Correlation analysis displayed that Ca, especially the Mg/Ca in fruits, could be used to predict whether an orchard will develop cork spot disorder; Mg and B were associated with the appearance of symptoms, and the contents of Zn, Fe, and Mg, as well as Mg/B and Zn/B, might be strongly tied to the formation of cork spots in pears. This research provides insights into the occurrence of pear cork spot disorder and clarifies the role of minerals.

**Keywords:** minerals; metabolites; cork spot disorder; Akizuki; pear

## 1. Introduction

In pear fruits, cork spot is a most common physiological disorder, except for hard end and water core [1,2], and has been found in a series of pear cultivars, such as 'Anjou', 'Alexander Lucas', 'Oushuu', 'Akizuki', and 'Chili' [3–6]. The typical characteristics of cork spot disorder are brown desiccated flesh beneath the fruit skin and slight pitting on the surface of fruit, which eventually result in a reduction of the market value of the fruits [3,7], which seriously restricts the development of the pear industry [4,8].

In order to find a solution to reduce or eliminate cork spot incidence, the regularities in the occurrence of cork spot disorder were identified. It has been reported that the cork spots appear on all parts of the fruit in Japanese pear (*P. pyrifolia* Nakai) 'Akizuki' and 'Oushuu' [5] and Chinese pear 'Chili' (*P. bretschneideri* Rehd.) [9]. But, in Cui's research, the cork spots were distributed close to the calyx end and outer flesh of 'Akizuki' (*P. pyrifolia* Nakai) fruits [10]. The onset time of this disorder is from the early developmental stage to the harvest time, and even after storage, depending on the cultivar. In 'Akizuki' pear, the

cork spot symptoms are usually observed more markedly in the late maturing fruit [3,5]. The severity of this disorder showed an association with the fruit fresh weights: more cork spots appeared on larger fruit [3,5]. In addition, the diameter of fruit pedicels, the calyx concave depth, and total soluble solids were reported to be changed in disordered fruits [3]. A wide range of environmental, physiological, and biochemical factors, such as tree age, soil conditions, the climate of the growing season [11], rootstocks [12], flower position and flowering time [13], growth regulators [14], as well as different daily orchard managements, including irrigation, fertilization, and pruning [8,15,16], have all been reported to influence the incidence of cork spot disorder. An imbalance in mineral nutrition in the fruit was regarded as the principal influence factor for cork spot disorder [17,18], of which calcium (Ca) deficiency has received the most attention [19,20]. For example, in 'Anjou' pear (*P. communis* L.) fruit, Ca was negatively correlated with cork spot both at harvest and after storage [15]. Foliar spraying of exogenous Ca2+ solution during the growing period could effectively reduce the incidence of cork spot disorder [3,21–23]. In addition, boron (B), magnesium (Mg), potassium (K), and nitrogen (N) contents and the ratio of them to Ca have also been reported to be associated with the incidence of cork spot disorder [6,8,15,17,20,24–27]. It was suggested that the interaction of Ca with other mineral elements may also affect the absorption of Ca by fruit [28,29]. However, recent studies have shown that Ca deficiency is not observed in the disordered fruit, but much higher levels of Ca ions are localized in the cell wall compared with normal fruits [3,4], and Ca spray could decrease the incidence of cork spot disorder in *P. pyrifolia* Nakai [3]. The relationships of minerals or the ratios of them to Ca with the incidence of cork spot disorder were not consistent in different studies. So, the relationship between mineral concentration and pear cork spot disorder is still not clear, and the mechanism underlying cork spot disorder occurrence remains unknown.

To date, cork spot disorder studies have been largely limited to the study of occurrence regularities and influence factors as mentioned above. The changes in metabolism related to cork spot disorder have not been studied in pear and the relationship between minerals and cork spot disorder is unclear. So, in our research, we adopted a widely targeted metabolome to understand the changes in metabolites in the disordered fruit and cork spot position and the correlation analysis of minerals and metabolites, which might allow further understanding of the occurrence of pear cork spot disorder and make clear the minerals' role.

## 2. Materials and Methods

### 2.1. Plant Materials

Pear fruits (*P. pyrifolia* Nakai) were obtained from two orchards with no cork spot disorder symptoms and more than 20% of fruits displaying cork spot disorder symptoms, respectively, in Pingdu, Shandong Province of China. The two orchards belong to one company with the same field management. In the two orchards, 7-year-old 'Akizuki' pear trees grafted on *P. betulaefolia* Bunge rootstocks and planted with spacing of 4 m × 3 m were selected for the experiment. Cork spotted and healthy fruits were collected at maturity, 130 days after full bloom (DAFB) in 2021 and 2022. Fleshes beneath the fruit skin dissected from healthy fruits were divided into high-quality fleshes (HQ, harvested in the orchard with no cork spotted fruits) and good-quality fleshes (G, harvested in the cork spot disorder orchard). Fleshes dissected from the cork spot disorder fruits were divided into normal fleshes (DN) and cork spotted fleshes (DS). A diagram of sampling is displayed in Figure 1. There were three replicates for each sample, and the fleshes of every five pears were mixed as one repeat. The samples were immediately frozen in liquid nitrogen and stored at −80 °C for further use.

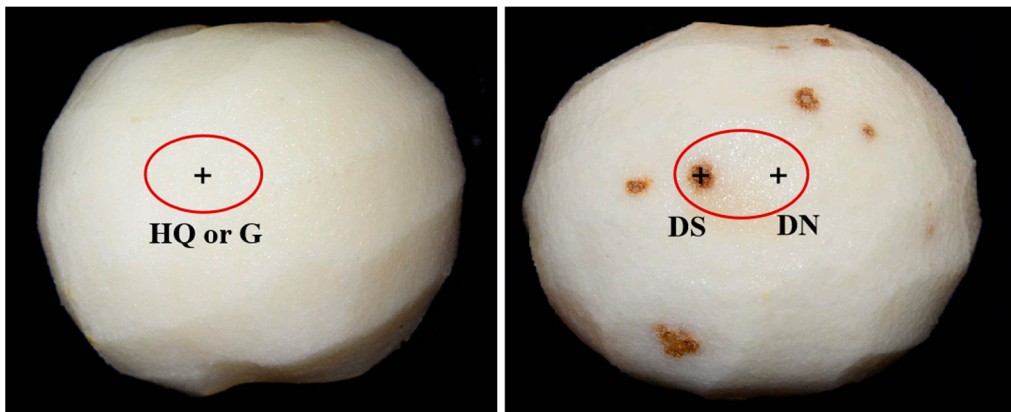

**Figure 1.** The diagram of sampling for HQ, G, DN, and DS. HQ, fleshes from healthy fruit in orchard where cork spot did not occur; G, fleshes from healthy fruit in cork spotted orchard; DN, normal fleshes from cork spotted fruit; DS, cork spotted fleshes.

### 2.2. Mineral Content Analysis

Fruit fleshes were dried in an oven at 105 °C for 30 min, followed with 75 °C treatment until a constant weight was achieved. Each 0.5 g sample was digested and dissolved with a mixture of 2 mL of perchloric acid and 10 mL of nitric acid. Then, the total contents of Ca, Mg, K, Fe, Zn, B, and P were analyzed by using an ICP-OES optima 8000 (PerkinElmer Inc., Waltham, MA). For the determination of total N content, 1 g of dried flesh sample was digested and dissolved with a mixture of 10 to 12 mL of concentrated $H_2SO_4$, 0.3 g of $CuSO_4$, and 3 g of $Na_2SO_4$. Then, the N content was assayed using a Kjeldahl nitrogen apparatus (K9860; Hanon instruments Co., Ltd., Jinan, China). The statistical analysis was carried out using SPSS ver. 17.0.

### 2.3. Enzyme Activity Analysis of POD, CAT, and SOD and Content Detection of MDA

The activity analysis of antioxidant enzymes was conducted following the instructions for SOD (G0103W, Grace Biotechnology, Suzhou, China), POD (G0107W, Grace Biotechnology, Suzhou, China), and CAT (G0105W, Grace Biotechnology, Suzhou, China) kits. The determination of MDA content was conducted following the instructions for an MDA content kit (G0109W, Grace Biotechnology, Suzhou, China).

### 2.4. Metabolite Extraction and Analysis

The freeze-dried samples were crushed using a mixer mill (MM 400, Retsch) with a zirconia bead for 1.5 min at 30 Hz. Then, 100 mg of powder was weighed and extracted overnight at 4 °C with 1.0 mL of 70% aqueous methanol. Following centrifugation at 10,000 g for 10 min, the supernatant was then extracted using an SPE cartridge. Then, the extracts were absorbed (CNWBOND Carbon-GCB SPE Cartridge, 250 mg, 3 mL; ANPEL, Shanghai, China) and filtrated (SCAA-104, 0.22 μm pore size; ANPEL, Shanghai, China) before LC-MS analysis. Quality control (QC) samples were mixed for all samples to measure the reproducibility of the whole experiment. Then, the compounds extracted were analyzed using an LC-ESI-MS/MS system (UPLC, Shim-pack UFLC SHIMADZU CBM30A; MS/MS, Applied Biosystems 6500 Q TRAP) by Gene denovo Biotechnology Co., Ltd. (Guangzhou, China). Two microliters of each sample was injected into a Waters ACQUITY UPLC HSS T3 C18 column (100 × 2.1 mm, 1.8 μm) operating at 40 °C and a flow rate of 0.4 mL/min. The mobile phases used were acidified water (0.04% acetic acid) (Phase A) and acidified acetonitrile (0.04% acetic acid) (Phase B). Compounds were separated using the following gradient: 95:5 Phase A/Phase B at 0 min; 5:95 Phase A/Phase B at 11.0 min; 5:95 Phase A/Phase B at 12.0 min; 95:5 Phase A/Phase B at 12.1 min; 95:5 Phase A/Phase B at 15.0 min. The effluent was connected to an ESI–triple quadrupole–linear ion trap (Q TRAP)–MS.

LIT and triple quadrupole (QQQ) scans were acquired on a triple quadrupole–linear ion trap mass spectrometer (Q TRAP), the AB Sciex QTRAP6500 System, equipped with an

ESI–turbo ion-spray interface, operating in positive ion mode and controlled by Analyst 1.6.1 software (AB Sciex). The operation parameters were as follows: ESI source temperature 500 °C; ion spray voltage (IS) 5500 V; curtain gas (CUR) 25 psi; the collision-activated dissociation (CAD) was set to the highest setting. QQQ scans were acquired as MRM experiments with optimized declustering potential (DP) and collision energy (CE) for each individual MRM transition. The *m/z* range was set between 50 and 1000.

### 2.5. Data Pre-Processing and Metabolite Identification

Data filtering, peak detection, alignment, and calculations were performed using Analyst 1.6.1 software. To produce a matrix containing fewer biased and redundant data, peaks were checked manually for signal/noise (s/n) > 10, and the redundant signals caused by different isotopes, in-source fragmentation, $K^+$, $Na^+$, and $NH_4^+$ adducts, and dimerization were removed. To facilitate the identification/annotation of metabolites, accurate *m/z* for each Q1 was obtained. Total ion chromatograms (TICs) and extracted ion chromatograms (EICs or XICs) of QC samples were exported to give an overview of metabolite profiles of all samples. The area of each chromatographic peak was calculated. Peaks were aligned across the different samples based on spectral pattern and retention time. Through searching the internal database of Gene denovo Biotechnology Co., Ltd. (most of the metabolites' information in the database was collected through standards, and some are manually identified by the literature) and public databases (MassBank, KNApSAcK, HMDB, MoTo DB, and METLIN) [30,31] and comparing the *m/z* values, the RT, and the fragmentation patterns with the standards, metabolites were identified. The fragmentation patterns are listed in Table S1. After the metabolites' identification, the metabolic pathway map was drawn with reference to the KEGG metabolic database (http://www.genome.jp/kegg/) (accessed on 10 March 2022).

### 2.6. Multivariate Statistical Analysis

For a preliminary visualization of differences between different groups of samples, an unsupervised dimensionality reduction method principal component analysis (PCA) was applied to all samples using R package models (http://www.r-project.org/) (accessed on 10 March 2022).

### 2.7. Differential Metabolite Analysis

A variable importance in projection (VIP) score of the OPLS model was applied to rank the metabolites that best distinguished two groups. The threshold of the VIP was set to 1. In addition, a *t*-test was also used as a univariate analysis for screening differential metabolites. Those metabolites with a *p*-value < 0.05 and VIP > 1 were considered to be differential metabolites between two groups. The heatmap was drawn with HemI software (version 1.0, http: /hemi.biocuckoo.org/) (accessed on 10 March 2022).

### 2.8. Statistical Analysis

The Venn diagrams and correlation analysis of minerals and metabolites were performed using a free online platform for data analysis (https://www.omicsolution.org/wkomics/main/) (accessed on 21 October 2022). The statistical analysis was carried out using SPSS ver. 17.0.

## 3. Results

### 3.1. Observation of the Cork Spot Symptoms at Harvest Time

At about 130 d after blooming, the fruits were harvested. As shown in Figure 2, compared with the healthy fruits (Figure 2d–f), the cork spotted ones displayed slight pitting on the surface (Figure 2a). In the anatomical observation of disordered fruits, we found that the cork spots were mainly distributed on the outer flesh beneath the fruit peel (Figure 2b). A few small cork spots were occasionally found in the inner flesh (Figure 2c).

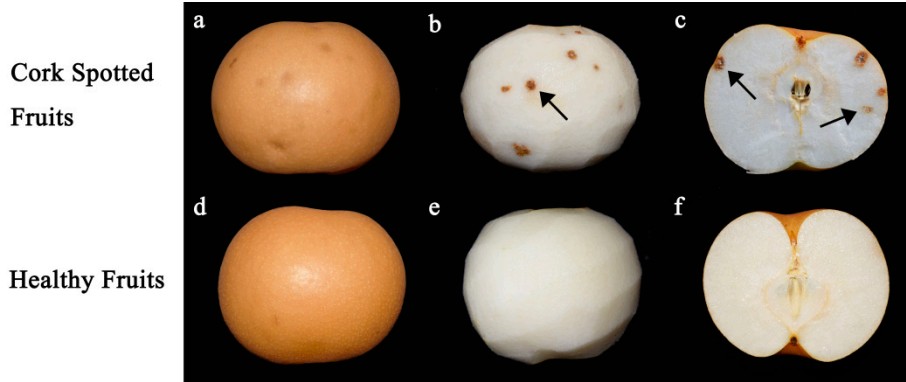

**Figure 2.** Observation of the cork spot symptoms at harvest time. (**a–c**) Cork spotted fruits; (**d–f**) healthy fruits; black arrows indicate the cork spotted tissues.

*3.2. Analysis of Mineral Element Contents among Different Fruit Fleshes*

To identify the relationship of mineral element contents and cork spot disorder, we first detected the contents of N, Ca, K, Mg, B, Cu, Fe, S, Zn, and P among HQ (the fleshes from the fruits harvested in the orchard with no cork spotted fruits), G (the fleshes from the healthy fruits harvested in the cork spot disorder orchard), DN (the normal fleshes from the fruits with cork spot disorder symptoms), and DS (the cork spotted fleshes from the fruits with cork spot disorder symptoms). As shown in Table 1, the content of N was higher in HQ and DS. The contents of Zn, Fe, Cu, S, and Mg were all highest in DS, the cork spot group. Compared with HQ and G, the Fe, Cu, and S contents were increased in DN, and that of Mg was lower in DN. The contents of B and Ca were highest in HQ, while the fruits in the orchard with cork spot disorder accumulated less B and Ca. The B content remained at a relatively high level in G, but was lower in cork spot fruits (DN and DS). Meanwhile, the Ca content was lower in G and DN and there was accumulation in DS. The highest contents of K and P were detected in G. The lowest content of K was detected in DS. The lowest content of P was detected in HQ. These results suggested that the cork spot disorder of pear might be related to the mineral contents.

*3.3. Detection of Antioxidant Enzyme Activities and MDA Contents*

The process of disease in plants is often accompanied by changes in cell membrane structure, leading to a loss of selective permeability. The accumulation of reactive oxygen species (ROS), membrane lipid peroxidation, and the accumulation of MDA are the main reasons for the loss of selective permeability. In order to identify whether membrane lipid peroxidation happens in cork spot disorder fruits, we detected the activities of antioxidant enzymes, including SOD, POD, and CAT, as well as the MDA contents. As shown in Figure 3a,b, no obvious differences in SOD and POD activities were detected among HQ, G, and DN, but those of DS were obviously higher than those of HQ, G, and DN. No differences in CAT activities were detected among HQ, G, DN, and DS (Figure 3c). The content of MDA in DS was also higher than those in HQ, G, and DN (Figure 3d). These results suggested that the cork spots were damaged flesh tissues which suffered stresses.

**Table 1.** Mineral element contents analysis of normal and cork spotted fleshes in healthy and cork spotted fruits.

| Samples | N (mg/kg) | Zn (mg/kg) | B (mg/kg) | Fe (mg/kg) | Cu (mg/kg) | Ca (mg/kg) | K (mg/kg) | Mg (mg/kg) | P (mg/kg) | S (mg/kg) |
|---------|-----------|------------|-----------|------------|------------|------------|-----------|------------|-----------|-----------|
| HQ | 3.19 ± 0.57 ab | 5.34 ± 0.87 b | 60.23 ± 9.41 a | 10.75 ± 1.36 d | 1.45 ± 0.25 c | 745.43 ± 78.03 a | 9401.12 ± 1323.53 bc | 580.75 ± 43.76 c | 761.51 ± 20.12 d | 496.95 ± 4.85 c |
| G | 2.02 ± 0.14 c | 4.80 ± 0.32 b | 49.80 ± 0.97 b | 21.45 ± 1.01 c | 1.19 ± 0.01 c | 194.91 ± 5.33 c | 12,995.33 ± 170.44 a | 678.39 ± 13.65 b | 1086.65 ± 23.38 a | 472.42 ± 31.23 cd |
| DN | 2.88 ± 0.06 b | 5.90 ± 0.93 b | 32.03 ± 0.56 c | 24.24 ± 1.14 b | 3.30 ± 0.05 b | 107.94 ± 8.97 d | 9932.20 ± 26.67 b | 478.84 ± 9.55 d | 885.68 ± 8.91 c | 712.39 ± 9.35 b |
| DS | 3.75 ± 0.29 a | 14.68 ± 1.12 a | 31.97 ± 0.60 c | 57.90 ± 8.91 a | 6.65 ± 0.05 a | 483.5 ± 11.26 b | 7381.43 ± 105.21 d | 1301.83 ± 19.90 a | 941.08 ± 7.39 b | 1141 ± 94.86 a |

Note: HQ, the fleshes of healthy fruits harvested in the orchard without cork spot disorder; G, the fleshes of healthy fruits harvested in the cork spot disorder orchard; DN, normal fleshes of cork spotted fruits; DS: cork spotted fleshes. Data are presented as means ± standard errors (SEs). Different letters in the same column indicate significant differences at the $p < 0.05$ level by Tukey's test.

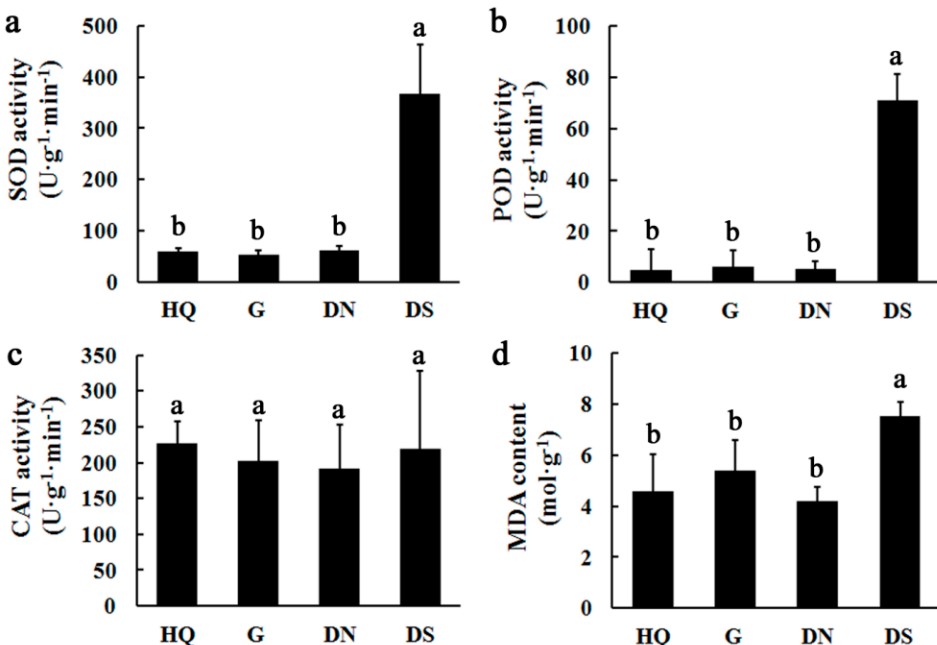

**Figure 3.** Activity analysis of SOD, POD, and CAT and detection of MDA content. (**a**) SOD activity, (**b**) POD activity, (**c**) CAT activity, (**d**) MDA content. Different letters in one graph indicate significant differences at the $p < 0.05$ level by Tukey's test.

### 3.4. Identification of Metabolites

Through broadly targeted metabolome analysis, a total of 1024 known metabolites were identified from all the samples (Table S1), involved in both the primary metabolism and the secondary metabolism pathways. Of all the compounds, flavonoids were the most common (214, 20.8%), followed by phenolic acids (205, 20%), lipids (142, 13.8%), amino acids (82, 8%), organic acids (78, 7.6%), saccharides and alcohols (72, 7%), nucleotides (49, 4.7%), terpenoids (46, 4.4%), lignans and coumarins (44, 4.2%), alkaloids (43, 4.1%), vitamins (18, 1.7%), tannins (17, 1.6%), and others (13, 1.3%) (Figure 4a). These compounds were involved in 99 pathways, including biosynthesis of amino acids, ABC transporters, carbon metabolism, flavonoid biosynthesis, phenylpropanoid biosynthesis, degradation of aromatic compounds, galactose metabolism, ascorbate and aldarate metabolism, and so on (Figure 4b).

Among the four samples, the highest number of identified metabolites (969) was detected in G, followed by HQ (960), DS (928), and DN with the fewest identified metabolites (859) (Figure 4c). Among the four sample tissues, 822 metabolites were detected in all the four sample tissues. Furthermore, 7, 10, 0, and 34 metabolites were detected only in HQ, G, DN, and DS samples, respectively. Among the 34 DS-specific metabolites, 10 terpenoids, 7 phenolic acids, and 7 flavonoids were detected. The specific metabolites in HQ and G were mainly phenolic acids and flavonoids. These results suggested that there might be biosynthesis of new metabolites, especially terpenoids, in DS, the cork spot group.

### 3.5. Metabolite Profiling Analysis and Differential Metabolite Identification

To examine the variation of the metabolite profiling, a PCA model was constructed for all the samples with a total variation of 67.6% explained by PC1 and PC2 (Figure 5a). The replicate samples scattered closely on the scoring plot, indicating the good stability in our detection (Figure 5a). For different samples, the three replicates of DS were grouped together, and the samples of HQ, G, and DN were clustered together, reflecting a big difference among the normal and cork spotted fleshes (Figure 5a).

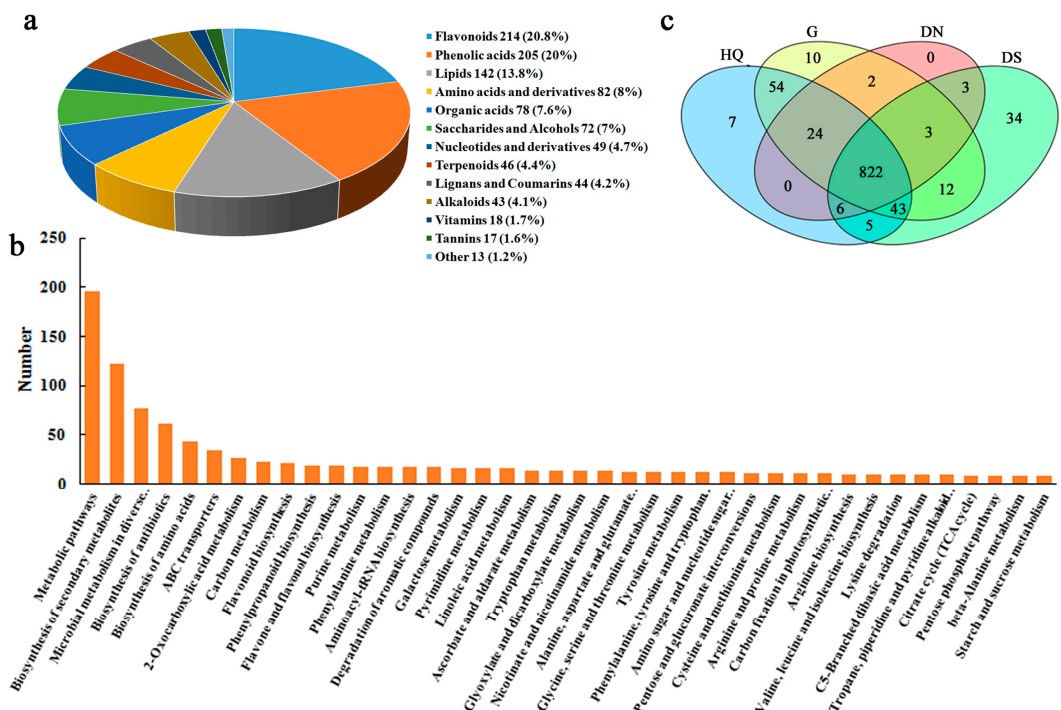

**Figure 4.** Analysis of the compounds isolated from pear fruits. (**a**) Component analysis of the isolated compounds in pear fruits; (**b**) the top 40 pathways through KEGG analysis of compounds isolated from pear fruits; (**c**) Venn analysis of compounds isolated from pear fruits.

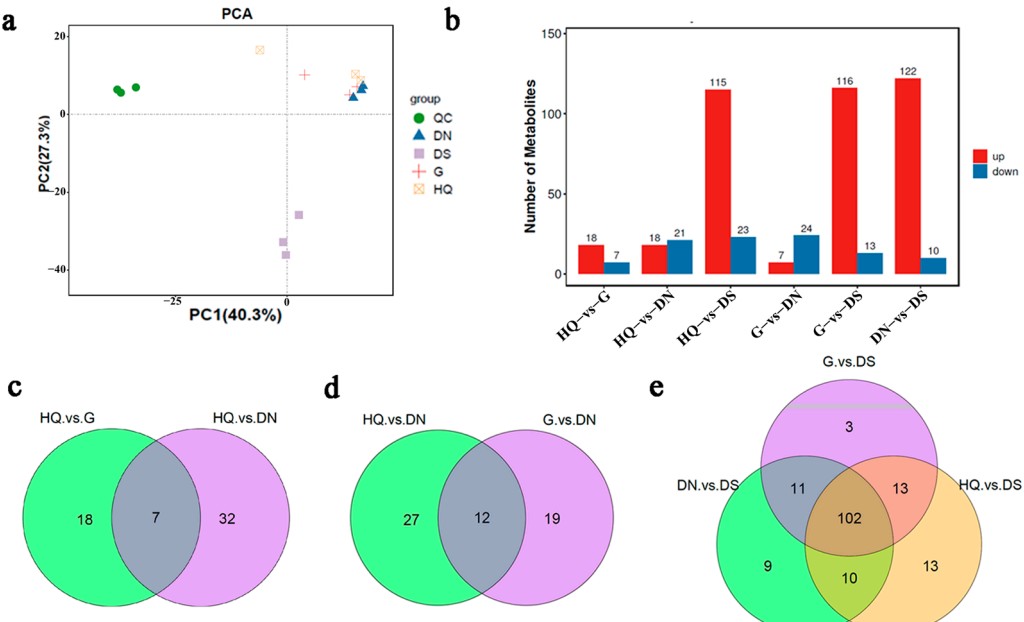

**Figure 5.** The PCA of all the samples and differential metabolite analysis in all samples. (**a**) The PCA score plot of all the samples according to their metabolite profiles; (**b**) the differential metabolites among different pear fruit fleshes; (**c**) Venn diagram showing the numbers of differential metabolites in HQ vs. G and HQ vs. DN groups; (**d**) Venn diagram showing the numbers of differential metabolites in HQ vs. DN and G vs. DN groups; (**e**) Venn diagram showing the numbers of differential metabolites in HQ vs. DS, G vs. DS, and DN vs. DS groups; QC, Quality Control.

To determine the metabolites that were critical among HQ, G, DN, and DS, the metabolite data were further analyzed by OPLS-DA. The OPLS-DA results showed that different samples clustered together. The values of Q2Y were all >0.5, and most close to 1, indicating

the reliability of the metabolomic data (Figure S1). VIP values are used to illustrate the importance of variables. When the VIP of a metabolite is >1, it indicates that the metabolite is important in separating different samples. So, we combined the VIP value of OPLS-DA and the *t*-test *p*-value to screen for significantly different metabolites between different comparison groups. The threshold of difference was as follows: VIP > 1 and *t*-test $p < 0.05$ in the OPLS-DA model.

Through the pairwise comparison, 25, 39, 138, 31, 129, and 132 differential metabolites were detected in 6 comparisons (HQ vs. G, HQ vs. DN, HQ vs. DS, G vs. DN, G vs. DS, and DN vs. DS) (Figure 5b). This result indicated that there were more changes in metabolism of the normal and cork spotted flesh tissues.

To analyze the differences of fruits in different orchards, we first identified the metabolism differences in fruits between different orchards with/without disordered fruits through comparing HQ with G and DN. Among HQ vs. G and HQ vs. DN, there were seven shared differential metabolites (Figure 5c), including four organic acids: aminomalonic acid, succinic acid, 3-ureidopropionic acid, and methylmalonic acid, two terpenoids: maslinic acid and 2α-hydroxyursolic acid, as well as one saccharide or alcohol: D-threonic acid (Table 2). Among these seven differential metabolites, the contents of organic acids and D-threonic acid were higher in HQ, and those of terpenoids were higher in G and DN (Table 2). These results suggested that the high-quality fruits in the orchard without cork spot disorder contained more organic acids and D-threonic acid, as well as fewer terpenoids, compared with the fruits in the orchard with cork spot disorder.

To further analyze the important differential metabolites related to cork spot disorder, we analyzed the differential metabolites in the groups of G vs. DN and HQ vs. DN. As shown in Figure 5d, there were 12 shared differential metabolites, including 3 saccharides and alcohols, 3 lipids, 2 organic acids, 2 terpenoids, and so on (Table 3). Among these 12 differential metabolites, the contents of 3 lipids (17-hydroxylinolenic acid, lysoPC 18:1(2n isomer), and lysoPC 18:1), 1 organic acid (L-pipecolic acid), 1 amino acid or derivative (cycloleucine), and 1 other (4-guanidinobutanal) were higher in HQ and G, while those of 1 organic acid (isocitric acid) and 3 saccharides and alcohols (D-sucrose, dulcitol, D-mannitol) were high in DN. It was interesting that the content of the two terpenoids (maslinic acid and 2α-hydroxyursolic acid) were higher in G, followed by DN and then HQ (Table 3). These results suggested that metabolism of lipids, organic acids, saccharides and alcohols, and terpenoids might change when cork spot disorder occurs in fruits.

Then, in order to identify the different metabolites of normal and cork spotted fleshes, we analyzed the differential metabolites in the groups of HQ vs. DS, G vs. DS, as well as DN vs. DS. As shown in Figure 5e, 138, 129, and 132 differential metabolites were obtained in the groups of HQ vs. DS, G vs. DS, and DN vs. DS. Among these metabolites, 102 were shared differential metabolites. To further analyze the differences in the 102 shared differential metabolites among normal and cork spotted pulps, hierarchical cluster analysis was performed (Figure 6). The differential metabolites in three biological replicates showed similar levels within the same sample tissue. Ninety-nine metabolites had much higher concentrations in DS, including 24 phenolic acids, 21 terpenoids, 17 flavonoids, 10 lignans and coumarins, 10 tannins, 9 lipids, and so on. Meanwhile, three metabolites (6-methylmercaptopurine, 2-decanol, and 1-decanol) displayed lower contents in DS (Table S2). These results suggested that, in cork spots, a lot of phenolic acids, terpenoids, and flavonoids were highly accumulated.

**Table 2.** Differential metabolites among healthy fleshes of fruits in different orchards.

| Name | Compound | Class | Log2 (G/HQ) | *p*-Value | VIP | log2 (DN/HQ) | *p*-Value | VIP |
|---|---|---|---|---|---|---|---|---|
| mws0470 | Methylmalonic acid | Organic acids | −1.13 | 0.000682108 | 14.90 | −1.05 | 0.00474035 | 12.73 |
| mws0192 | Succinic acid | Organic acids | −1.16 | 0.000609659 | 15.13 | −1.09 | 0.004809102 | 13.01 |
| pme3096 | Aminomalonic acid | Organic acids | −1.21 | 0.000230108 | 2.95 | −1.12 | 0.002645329 | 2.52 |
| Lmgn000160 | 3-Ureidopropionic acid | Organic acids | −0.48 | 0.014183107 | 1.94 | −1.01 | 0.005539138 | 2.39 |
| Lmzn006284 | 2α-Hydroxyursolic acid | Triterpenes | 4.21 | 0.008486822 | 3.21 | 2.09 | 0.002040357 | 1.27 |
| mws1610 | Maslinic acid | Triterpenes | 4.15 | 0.011787435 | 3.18 | 1.97 | 0.005822207 | 1.22 |
| mws0889 | D-Threonic acid | Saccharides and alcohols | −0.76 | 0.004782529 | 1.66 | −0.96 | 0.000892064 | 1.64 |

Note: Compounds selected as significantly different metabolites by VIP > 1 and *p*-value < 0.05.

**Table 3.** Differential metabolites among healthy and disordered fruits.

| Name | Compound | Class | Log2 (DN/HQ) | *p*-Value | VIP | Log2 (DN/G) | *p*-Value | VIP |
|---|---|---|---|---|---|---|---|---|
| pme2237 | Dulcitol | Saccharides and alcohols | 0.47 | 0.004784837 | 5.77 | 0.64 | 0.001583311 | 9.10 |
| pme0519 | D-Sucrose | Saccharides and alcohols | 0.81 | 0.013778942 | 2.37 | 0.55 | 0.045412344 | 2.66 |
| mws1155 | D-Mannitol | Saccharides and alcohols | 0.67 | 0.015519951 | 3.89 | 0.84 | 0.011389627 | 6.04 |
| pmp001281 | LysoPC 18:1 | Lipids | −0.98 | 0.015533774 | 1.69 | −0.41 | 0.008074391 | 1.34 |
| Lmhp010190 | LysoPC 18:1(2n isomer) | Lipids | −0.97 | 0.034051831 | 1.64 | −0.48 | 0.016473192 | 1.45 |
| Zmyn004676 | 17-Hydroxylinolenic acid | Lipids | −1.25 | 0.01554908 | 1.02 | −0.90 | 0.034279247 | 1.11 |
| Zmyn000453 | Isocitric acid | Organic acids | 0.46 | 0.002719251 | 3.27 | 0.50 | 0.035685538 | 4.03 |
| MWS0811 | L-Pipecolic acid | Organic acids | −1.21 | 0.031421359 | 2.30 | −0.97 | 0.014435477 | 2.87 |
| mws1610 | Maslinic acid | Triterpenes | 1.97 | 0.005822207 | 1.22 | −2.19 | 0.022651996 | 3.22 |
| Lmzn006284 | 2α-Hydroxyursolic acid | Triterpenes | 2.09 | 0.002040357 | 1.27 | −2.12 | 0.017226549 | 3.24 |
| ML10181668 | Cycloleucine | Amino acids and derivatives | −1.15 | 0.008737324 | 2.88 | −0.70 | 0.003361136 | 2.90 |
| Zmdp000376 | 4-Guanidinobutanal | Others | −1.21 | 0.017358602 | 2.81 | −0.74 | 0.001701593 | 2.87 |

Note: Compounds selected as significantly different metabolites by VIP > 1 and *p*-value < 0.05.

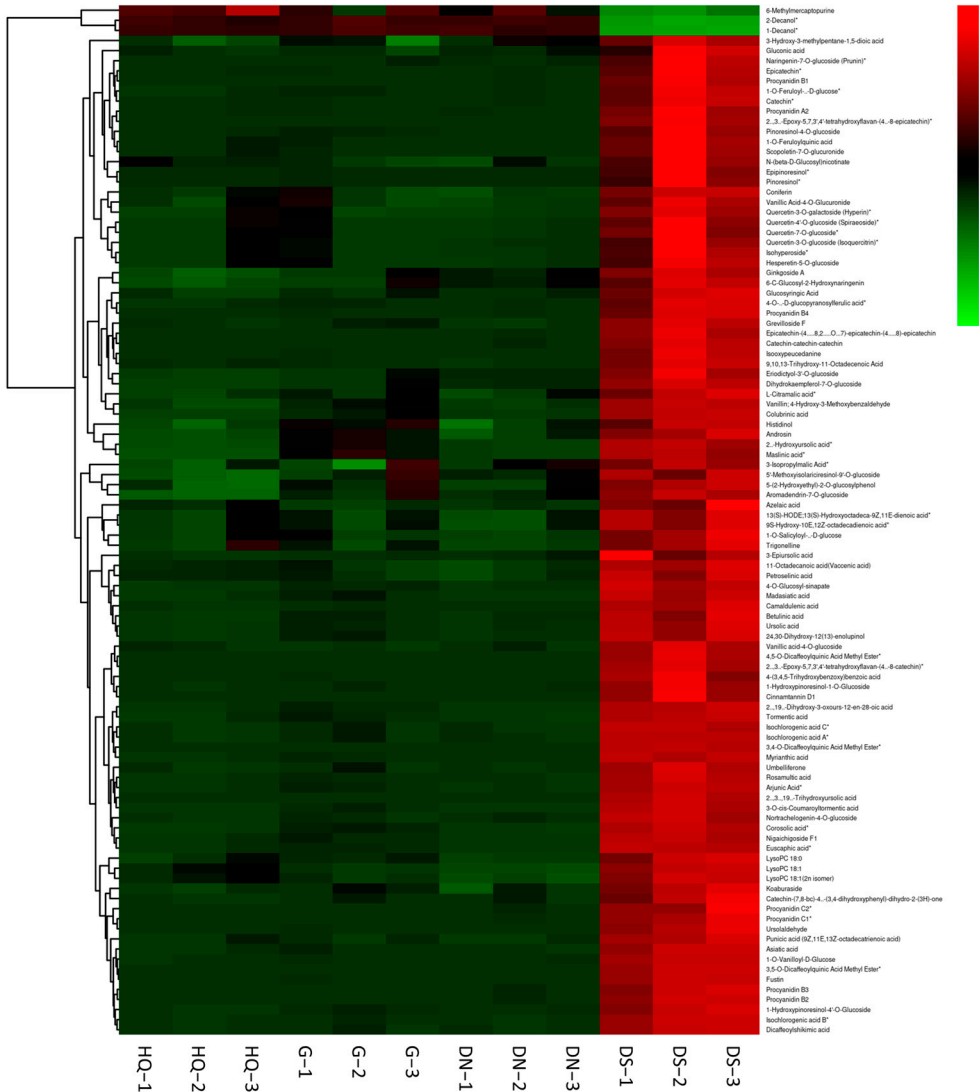

**Figure 6.** Hierarchical clustering of the differential metabolites in the groups of HQ vs. DS, G vs. DS, and DN vs. DS. Each column represents a biological replicate. * mean isomeride.

*3.6. Correlation Analysis of Mineral Elements and Differential Metabolites*

Pearson correlation analysis was conducted on the mineral element contents and differential metabolites mentioned above. First, we analyzed the correlation of minerals with the seven differential metabolites which represented the differences between orchards with or without cork spot disorder. As shown in Figure 7a, Ca displayed a positive correlation with three organic acids, including methylmalonic acid (mws0470), succinic acid (mws0192), and aminomalonic acid (pme3096). Mg/Ca had a negative correlation with five of the seven metabolites, methylmalonic acid (mws0470), succinic acid (mws0192), aminomalonic acid (pme3096), D-threonic acid (mws0889), and 3-ureidopropionic acid (Lmgn000160). So, we speculated that the content of Ca, especially the Mg/Ca in fruits, might be used to predict whether an orchard will develop cork spot disorder.

Then, we analyzed the correlation of minerals with the 12 differential metabolites which represented the differences between healthy and disordered fruits. As shown in Figure 7b showed positive correlation with one organic acid, l-pipecolic acid (MWS0811), as well as cycloleucine (ML10181668) and 4-guanidinobutanal (Zmdp000376). And Mg had positive correlation with two triterpenes, 2α-jydroxyursolic acid (Lmzn006284) and maslinic acid (mws1610), as well as three lipids, lysoPC 18:1 (pmp001281), lysoPC 18:1(2n isomer) (Lmhp010190), and 17-hydroxylinolenic acid (Zmyn004676). So we speculated that

Mg and B might be associated with the appearance of symptoms of cork spot disorder of pear.

Lastly, we analyzed the correlation of minerals with the 102 differential metabolites which represented the differences between healthy and cork spot fleshes. As shown in Figure 8, Zn, Fe, Mg, Mg/B, and Zn/B showed positive correlations with most of the differential metabolites, mainly flavonoids, phenolic acids and terpenoids. These results suggested that the contents of Zn, Fe, and Mg, as well as Mg/B and Zn/B, might be strongly tied to the formation of cork spot in pears.

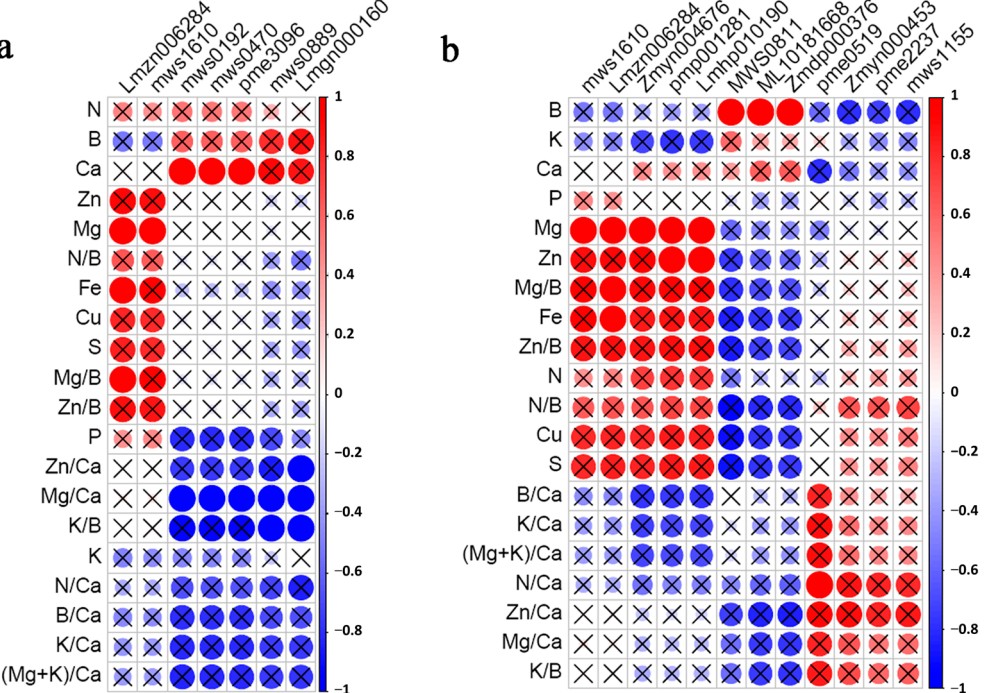

**Figure 7.** Correlation analysis between mineral element data and differential metabolites among healthy fleshes of fruits in different orchards (**a**) and among healthy and disordered fruits; (**b**) circles with a cross inside indicate no significant correlation at 0.05 level.

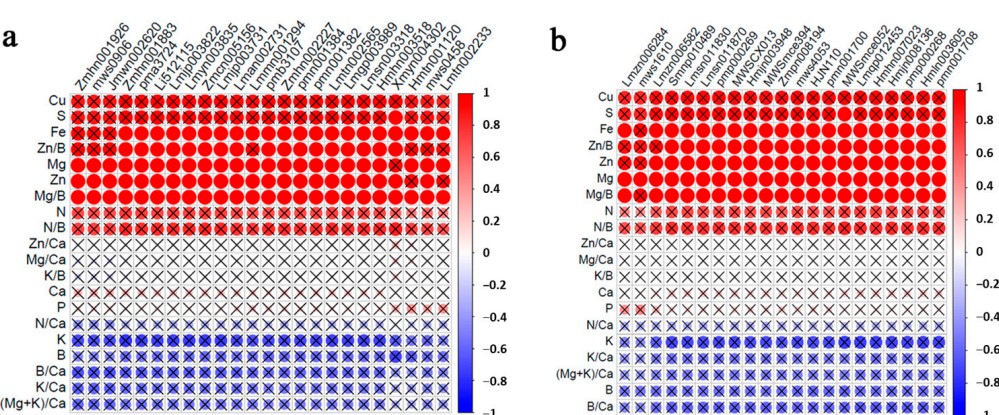

**Figure 8.** *Cont*.

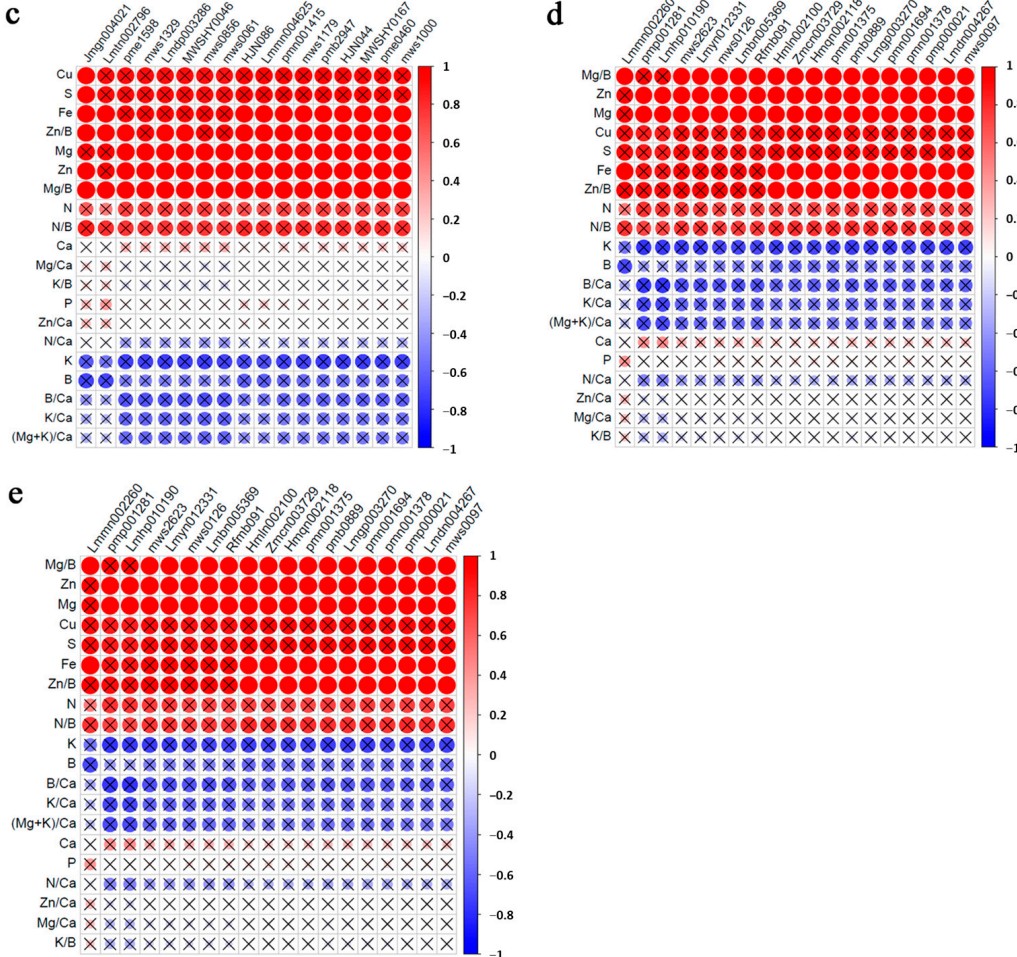

**Figure 8.** Correlation analysis between mineral element data and differential metabolites among normal and cork spotted fleshes. (**a**) Correlation analysis between mineral element data and differential phenolic acids; (**b**) correlation analysis between mineral element data and differential terpenoids; (**c**) correlation analysis between mineral element data and differential flavonoids; (**d**) correlation analysis between mineral element data and differential lignans and coumarins and lipids; (**e**) correlation analysis between mineral element data and other differential compounds. Circles with a cross inside indicate no significant correlation at 0.05 level.

## 4. Discussion

'Akizuki', a Japanese pear, is a very popular cultivar in China with large fruit size, pretty fruit shape, and pleasant taste. However, 'Akizuki' is very susceptible to cork spot disorder, which has caused great economic loss in the pear industry and has become a major obstacle to the continuing expansion of the cultivated area [3]. However, the mechanisms of cork spot disorder remain poorly understood. Therefore, the management techniques to prevent this disorder are often inadequate.

Cork spot disorder is a physiological disease. In has been reported that some diseases in plants are accompanied by membrane lipid peroxidation and loss of selective permeability. Through activity analysis of antioxidant enzymes and MDA content, we suggest that cork spots were damaged pulp tissues which suffered stresses (Figure 3). It is generally acknowledged that, under stress conditions, the balance between production and elimination of ROS is disturbed [32–35]. ROS take part in the process of programmed cell death (PCD) directly or act as a molecular signal in plants [36–40]. So, the cork spots may be composed of programmed apoptotic cells suffering stresses. In 'Akizuki' fruit, flesh tissue damage was found to occur prior to the development of cork spots through the X-ray CT scanning technique [41]. We hypothesized that some stresses induced the production of

a lot of ROS in some fruit flesh cells. Although the activities of SOD and POD increased in these cells, ROS was not completely cleared away for some reason. Then, ROS promoted programmed cell death, resulting in the tissue damage.

As a physiological disorder, cork spot disorder could lead to changes in metabolic bioprocesses. It was expected that cork spot disorder would be characterized by extensive metabolic changes in multiple pathways. Recent technical advancements in metabolome analyses have provided effective ways to identify metabolites and to elucidate complex metabolic bioprocesses in plants [42–45]. The fruit metabolome has been widely surveyed for physiological disorders in pear [2], apple [46], and so on. But, there is still no report of metabolome analysis on cork spot disorder in pear. In our research, we analyzed the metabolome related to cork spot disorder in 'Akizuki'. We identified 1024 known metabolites. Among these metabolites, flavonoids and phenolic acids accounted for more than 40% (Figure 4a). In the embryos of pear cultivar 'Cuiguan' (*P. pyrifolia* (Burm.f.) Nakai), the number of flavonoids and phenolic acids was also the highest, accounting for 35% [44]. These metabolites can exert antioxidant activity on fruits, and even on humans after fruit consumption, allowing them to act as efficient scavengers of reactive oxygen species (ROS).

Through analyzing the metabolite differences among HQ, G, DN, and DS, we identified the different metabolites among different fruits from different orchards, including healthy and disordered fruits, as well as healthy and cork spotted fleshes. Compared with the fruits in the orchard with cork spot disorder, more organic acids and D-threonic acid and fewer terpenoids were detected in high-quality fruits in the orchard without cork spot disorder (Table 2). Among healthy and disordered fruits, the contents of lipids, organic acid, saccharides and alcohols, and terpenoids displayed differences. These results suggested that these differential metabolites were related to cork spot disorder in fruits and might be used for prediction of cork spot disorder. Concerning the differences in the contents of saccharides and alcohols and organic acids, we also detected the soluble solids content in disordered and healthy fruits, with no obvious differences. So, the soluble solids index could not separate the disordered and the healthy fruits at harvest time, as suggested by previous reports [4,24,47]. Whether other metabolites could be used for disorder prediction still needs further study.

At the same time, we found that most of the differential metabolites were found among normal and cork spotted fleshes. Ninety-nine metabolites accumulated highly in DS, the cork spot group, including 24 phenolic acids, 21 terpenoids, 17 flavonoids, and so on (Figure 6). Concerning the activity analysis of antioxidant enzymes and MDA content, we suggest that cork spots were damaged pulp tissues which suffered stresses (Figure 3). In fact, plants have evolved diverse enzyme-based (including SOD, CAT, POD, etc.) and non-enzyme-based (such as phenolics, flavonoids, ascorbic acid (AsA), glutathione (GSH), etc.) antioxidant mechanisms to maintain the cellular ROS balance [48]. Triterpenoids are compounds with a carbon skeleton based on six isoprene units. Most triterpenoids are considered as defensive chemicals against pathogens in plants [49–51]. So, the accumulation of phenolic acids, flavonoids, and triterpenoids in cork spots might be a non-enzyme-based antioxidant pathway, which plays a role in clearing away the ROS. It is worth noting that we identified 46 terpenoids in total (Figure 4a), and close to 1/2 (21) were highly accumulated in DS. Combining the previous result that 10 terpenoids were DS-specific metabolites, we suggest that the high accumulation of terpenoids could be associated strongly with cork spot incidence, which needs further research.

Cork spot is a physiological disorder of fruit which occurs mostly in pears and apples. Many studies have shown a relationship between mineral composition and the development of disorders in fruits. A number of physiological disorders in apple and pear have been reported to be associated with low Ca content [52]. So, like other physiological disorders such as bitter pit of apples [2,53] and hard end and superficial scalding of pears [1,54], cork spot disorder in pear was generally believed to be a $Ca^{2+}$-deficiency-related physiological disorder [6,15,20–23,25]. At the same time, the N/Ca ratio in the fruit was shown to be closely correlated with bitter pit in apple [55]. In apple, boron (B) has also been reported

to play a fundamental role in physiological disorders through affecting basic structural features and physiological processes in plants [56,57]. It has been reported that B spray increased the B supply and reduced corky core development in apple. In 'Akibae' pear, B application can reduce the occurrence of water core [58]. Thus, nutritional control such as Ca and B spraying is the main means of reducing or preventing physiological disorders in apple or pear. Of course, this method may be species specific. It has been reported that in 'Kurenainoyume' apples, spraying calcium (Ca), boron (B), or both on the tree could not decrease the incidence of cork spot [24]. In our results, the contents of B and Ca were highest in HQ and lower in the fruits from the orchard with cork spot disorder. The contents of Zn, Fe, Cu, S, and Mg were all highest in DS, the cork spot group (Table 1). According to these results, we could speculate that the deficiency of B and Ca could cause the incidence of cork spot and the accumulation of Zn, Fe, Cu, S, and Mg was correlated with the formation of spots.

In order to verify our speculation, the correlations among the mineral contents and metabolites were analyzed. We found that the contents of Ca, especially the Mg/Ca in fruits, was associated with most of the differential metabolites between the two orchards (Figure 7a). Combined with the results that the Ca content was highest in HQ and lower in the fruits from the orchard with cork spot disorder (Table 1), we suggest that the deficiency of Ca content might lead to a higher incidence of cork spot disorder, and the incidence of cork spot disorder could be predicted from the Mg/Ca. At the same time, through analysis of the correlation of the mineral contents and differential metabolites between healthy and disordered fruits, we found that Mg, Zn, and B might be associated with most of those differential metabolites. Considering the content of B in DN was lower than HQ and G, we suggest that the deficiency of B in fruits led to the occurrence of cork spot symptoms. Meanwhile, the contents of Zn, Fe, and Mg were strongly tied to the differential metabolites between DS and healthy fleshes. As a result of the higher contents of Zn, Fe, and Mg in DS than healthy fleshes, we suggest that Zn, Fe, and Mg were easily accumulated in cork spot position.

In general, the deficiency of Ca or the increase in Mg/Ca in fruits of one orchard might lead to a higher incidence of cork spot disorder. The Ca and Mg/Ca contents could be used for prediction of cork spot disorder. It has been reported that Ca plays roles in stabilizing the cell membranes, cell walls, and extracellular pectinaceous bonding materials [59–61]. So, we speculated that insufficient Ca content would lead to poor stability of cell membranes, resulting in cell damage by external or internal factors. But, we also know that, in orchards with cork spot disorder, some fruits display cork spots while others do not. So, we speculated that simple Ca deficiency will not result in cork spots in pear. Could other factors lead to cork spot symptoms? Through our research, we suggest that on the basis of Ca deficiency, the deficiency of B in fruit could lead to cork spot symptoms. Although the function of B in apple and pear remains unknown, in higher plants, B plays important roles in numerous physiological processes, such as cell wall synthesis and lignification, cell wall structure maintenance, respiration, metabolism of carbohydrate or hormones, and membrane transport [62]. So, the deficiency of B might lead to the breakdown of cell walls of the pear fruits. It has been reported that Mg competes with Ca for binding sites on the cell membrane and may replace Ca on the surface of the plasma membrane but cannot replace its role in cell membrane structure and function [63]. So, excess Mg exacerbated the deficiency of Ca, causing the poor stability of cell membranes which were easily damaged by external or internal factors. What is the result of the imbalanced distribution of minerals? Maybe the necrosis and browning of vascular tissues, which have been observed in cork spotted 'Chili' [4].

Combining the results and previous reports, we give a description of the formation of cork spot disorder. During the fruit development, a series of physiological and biochemical factors, such as tree age, soil conditions, climate, rootstock, as well as different orchard management techniques, including irrigation, fertilization, and so on, may result in water and mineral transport differences. The distribution of some minerals, such as Ca, Mg,

Zn etc., displays imbalance among different cell masses, which affects the stability of cell membranes and cell wall structure. Some cells suffer water or nutrition deficits or other physiological stresses, which induce the production of ROS, resulting in membrane lipid peroxidation. In order to reduce membrane lipid peroxidation damage, the activities of SOD and POD increase, and a series of phenolic acids, flavonoids, and triterpenoids accumulate in the damaged area. But, as a result of physiological changes and instability in cell membranes and cell walls, ROS cannot be cleared away, leading to programmed cell death and further tissue damage and cork spot symptoms.

## 5. Conclusions

Through antioxidant enzyme activity, metabolome, and mineral content analysis on healthy and cork spot disordered 'Akizuki' fruits, we support that cork spot disorder of pear is related to the mineral contents, and the cork spots were damaged flesh tissues. A total of 1024 known metabolites were identified from all the samples and changes in metabolism were detected between normal and cork spotted flesh tissues. Correlation analysis displayed that Ca, especially the Mg/Ca in fruits, could be used to predict whether an orchard will develop cork spot disorder; Mg and B were associated with the appearance of symptoms, and the contents of Zn, Fe, and Mg, as well as Mg/B and Zn/B, might be strongly tied to the formation of cork spots in pears.

**Supplementary Materials:** The following supporting information can be downloaded at: https://www.mdpi.com/article/10.3390/horticulturae9070818/s1, Table S1: The 1024 known metabolites identified from all pear samples; Table S2: Differential metabolites among normal and cork spotted tissues; Figure S1: A supervised clustering analysis of different samples on their metabolite profiling.

**Author Contributions:** Y.Y. analyzed the data and wrote the manuscript. Y.Z. and A.L. conducted the experiments. P.J. collected the samples and participated in the experiments. Z.Q., Y.C. and X.Z. participated in data analysis. R.W., C.M., J.S., Z.C. and J.L. designed the experiments and reviewed the manuscript. Y.N. managed the orchard and provided pear samples. D.L. conceived and designed the experiments and provided the project administration and funding acquisition. All authors have read and agreed to the published version of the manuscript.

**Funding:** This research was supported by the Agricultural Variety Improvement Project of Shandong Province (2022LZGC011), the earmarked fund for CARS (CARS-28-07), the National Key Research and Development Program of China (2019YFD1001404), and the Science & Technology Specific Projects in Agricultural High-tech Industrial Demonstration Area of the Yellow River Delta (2022SZX30).

**Institutional Review Board Statement:** Not applicable.

**Informed Consent Statement:** Not applicable.

**Data Availability Statement:** The datasets used and/or analyzed during the current study are available from the corresponding author on reasonable request.

**Acknowledgments:** Not applicable.

**Conflicts of Interest:** The authors declare no conflict of interest.

## Abbreviations

SOD, superoxide; POD, peroxidase; MDA, malondialdehyde; Ca, calcium; B, boron; Mg, magnesium; K, potassium; N, nitrogen; DAFB, days after full bloom; HQ, the fleshes from the fruits harvested in the orchard with no cork spotted fruits; G, the fleshes from the healthy fruits harvested in the cork spot disorder orchard; DN, the normal fleshes from the fruits with cork spot disorder symptoms; DS, the cork spotted fleshes from the fruits with cork spot disorder symptoms; ROS, reactive oxygen species; AsA, ascorbic acid.

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
