# Peer review of "Mineral and Metabolome Analyses Provide Insights into the Cork Spot Disorder on ‘Akizuki’ Pear Fruit"

_horticulturae, doi:10.3390/horticulturae9070818_

Round 1

Reviewer 1 Report

This is an excellent study on cork spot disorder, one of the most important fruit disorders of Japanese pear, in which metabolomic analysis was performed along with mineral analysis. However, the discussion of the metabolome analysis are lacking; the relationship between the detected metabolites and the process of cork spot formation needs to be discussed as much as possible. 

Since the novelty of this paper lies in the metabolome analysis, the discussion needs a description that gives perspective to the metabolites; L389-407 alone is not sufficient. What exactly is the ‘some stress’ that the authors describe(L396) ? Based on the measurements of SOD and POD activity, why are they in an oxidative state? It is necessary to describe the steps from the occurrence of some stress to the formation of the physiological disorder ‘cork spot (lignified tissue)’. It would be better if the paper referring to ‘programmed cell death (Cui et al. 2021)’ could be cited and discussed. I look forward to further discussion description.

L4: ;→ ,

L88:    d → days

L90:  I don't know what ‘quality’ means and it is difficult to understand why each sample is abbreviated as 'HQ' and 'G' respectively.

L104:  The supernatant was then extracted using SPE cartridge

L111: (2.1mm*100mm, 1.8μm)→(100×2.1mm, 1.8μm)

L146:  T-test → t-test

L148:  T-test → t-test

L152:  List in the same order as the order of the results.

L155:  perchloric  → perchloric acids

L162:  List in the same order as the order of the results.

L165:  separately → respectively

L169:  veen → venn

L221:    Isn't it Tukey's test, not Student’s-t-test?  

L237:    Isn't it Tukey's test, not Student’s-t-test? 

L244:    Terpenoids → terpenoids

L247:    Phenylpropanoid → phenylpropanoid

L254:  separately → respectively

Figure 4:  Make the text in the figures larger and easier to read.

L262:  veen → venn

Figure 5a:  Make the text in the figures larger and easier to read.

Figure 5(c), (d), (e) : veen → venn

L280:  metabonomic  → metabolomic

L285:  T-test → t-test

Table2:  Reduce the size of the numerical values so that they do not have two columns.

Table2:  Vip → VIP

Table2:  Make the size of the P-value the same as the size of the other characters.

L308:  1 organic acid→ 2 organic acids

Table3:  Reduce the size of the numerical values so that they do not have two columns.

Table3:  Vip → VIP

Figure 6:  Make the text in the figures larger and easier to read.

Figure 7:  State the crosses a little more clearly so that it is easier to distinguish them from those without crosses.

Figure 8:  Figure is too small to see. Make it bigger.

L405:  Fig.3c → Fig.5e

L469:  d → days

References: References should be listed according to the format prescribed by the journal.

L512:  Delete 'Pyrus pyrifolia'.

L513:  Akizuki  → Akituki

L540: Provide the author's name correctly.

L586-590: These two references are cited?

The English text needs to be further brushed up.

Reviewer 2 Report

In this paper, the authors investigated the cork spot characteristics in Japanese pear 'Akizuki' (P. pyrifolia Nakai) through metabolome and mineral content analysis of healthy and disordered fruits. The authors observed increased activities of Superoxide (SOD) and Peroxidase (POD), as well as elevated malondialdehyde (MDA) content in cork spotted tissues. Furthermore, they identified a total of 1024 known metabolites and detected significant changes in metabolism between normal and cork spotted flesh tissues. The correlation analysis indicated the potential use of Ca, Mg/Ca ratio, Mg, B, Zn, Fe, Mg/B, and Zn/B in predicting and understanding cork spot disorder formation in pears.

The work presented in this paper is a significant contribution to the field and is commendably well-organized and well-written. However, there are some major and minor comments that should be addressed.

Major Comment:

The authors state that "A total of 1024 known metabolites were identified from all the samples, and more changes in metabolism were detected between normal and cork spotted flesh tissues." While this is an important finding, there is a discrepancy in the method section for data pre-processing and metabolite identification. The authors mention searching an internal database and public databases, comparing m/z values, retention time (RT), and fragmentation patterns with standards to identify metabolites. This raises a concern that the authors may have putatively identified the metabolites by searching the database with some best candidates, rather than confidently identifying them by standards. It is essential to provide a more detailed explanation of how the authors identified the 1024 metabolites in the method section. Additionally, including a comprehensive list of the 1024 detected metabolites with their m/z values, RT, and fragmentation patterns in the supplementary materials would greatly enhance the paper's significance and potential for the field.

Minor Comments:

In line 337-340, there are various combinations of letters and numbers following each metabolite, such as "methylmalonic acid (mws0470), succinic acid (mws0192), and aminomalonic acid (pme3096)." Please clarify the meaning of these combinations. If they are merely metabolic feature names, it is suggested to remove them in the resubmission. Additionally, for important metabolites used for correlation ananlaysis to draw important conclusions, such as methylmalonic acid, succinic acid,  aminomalonic acid, D-threonic acid, and 3- ureidopropionic acid, it is crucial to confidently identify them using authentic standards and provide detailed information in the supplementary materials.

The discussion section of the manuscript should be improved. The discussion is a vital part of a paper that should not only summarize key results but also highlight the insights and applicability of the findings for future research. Addressing important questions, such as the research gaps the study helps to address, the beneficiaries of the improvements, and future directions, will strengthen the discussion. Focusing the discussion would also enable the authors to state the contribution of the paper more clearly.

Finally, it is strongly recommended to include a separate conclusion section in the manuscript when resubmitting. The conclusion section serves as a vital component of a scientific paper, providing a concise summary of the key findings and their implications.

Round 2

Reviewer 2 Report

The revised manuscript has undergone significant improvements and is now suitable for acceptance in its present form.